# Mapping and Screening of Candidate Gene Regulating the Biomass Yield of Sorghum (*Sorghum bicolor* L.)

**DOI:** 10.3390/ijms25020796

**Published:** 2024-01-08

**Authors:** Mao Li, Qizhe Cai, Yinpei Liang, Yaofei Zhao, Yaoshan Hao, Yingying Qin, Xinrui Qiao, Yuanhuai Han, Hongying Li

**Affiliations:** 1Shanxi Key Laboratory of Minor Crops Germplasm Innovation and Molecular Breeding, Shanxi Agricultural University, Taigu, Jinzhong 030800, China; limao28ban@sxau.edu.cn (M.L.); 20200101105@stu.sxau.edu.cn (Q.C.); lyp@sxau.edu.cn (Y.L.); yfzhao@sxau.edu.cn (Y.Z.); haoyaoshan@sxau.edu.cn (Y.H.); qingyy@sxau.edu.cn (Y.Q.); 2College of Life Sciences, Shanxi Agricultural University, Taigu, Jinzhong 030800, China; 20211311910@stu.sxau.edu.cn; 3College of Agriculture, Shanxi Agricultural University, Taigu, Jinzhong 030800, China

**Keywords:** sorghum, biomass yield, leaf morphology, *sblob*, melatonin

## Abstract

Biomass yield is one of the important traits of sorghum, which is greatly affected by leaf morphology. In this study, a lobed-leaf mutant (*sblob*) was screened and identified, and its F2 inbred segregating line was constructed. Subsequently, MutMap and whole-genome sequencing were employed to identify the candidate gene (*sblob1*), the locus of which is Sobic.003G010300. Pfam and homologous analysis indicated that *sblob1* encodes a Cytochrome P450 protein and plays a crucial role in the plant serotonin/melatonin biosynthesis pathway. Structural and functional changes in the *sblob1* protein were elucidated. Hormone measurements revealed that *sblob1* regulates both leaf morphology and sorghum biomass through regulation of the melatonin metabolic pathway. These findings provide valuable insights for further research and the enhancement of breeding programs, emphasizing the potential to optimize biomass yield in sorghum cultivation.

## 1. Introduction

Sorghum (*Sorghum bicolor* L.) is the world’s fifth-largest cereal crop and a staple food for more than 500 million people, particularly in arid and semi-arid regions, where it exhibits moderate resistance to many stressors such as drought, salinity and bareness [1,2]. Sorghum is also a potential source of bioenergy, possessing excellent traits such as high biomass yield and a short growth period [3]. Sorghum’s high biomass yield translates into a substantial quantity of plant material, offering abundant feedstock for biofuel production [4]. For example, sweet-stemmed sorghum is a valuable resource when it comes to extracting sugars [5], while its leaves and stems contribute to the production of cellulose—a key component in biofuel and bioethanol manufacturing processes [1,4,5]. Therefore, the biomass yield of sorghum has important potential in energy production, in addition to economic viability and diverse agricultural applications [6]. It is particularly suitable for functional genetic studies related to various agronomic traits, such as leaf morphology, because it is characterized by 2n = 20 chromosomes as a diploid species [7].

Leaf morphology is one of the key agronomic traits affecting the biomass yield of sorghum, presenting a broad range of phenotypic variations relevant to its use as a potential energy crop [8]. It can also be used to determine the efficiency of photosynthesis, plant architecture and nutrient distribution [9,10], as larger leaf areas have been positively correlated with biomass yield characteristics in sorghum [11]. Moreover, leaf morphology can also affect nutrient uptake, which is crucial to the plant’s growth and biomass yield [12]. Although leaf morphology and biomass yield are important agronomic traits, the molecular genetic mechanism that controls these traits in sorghum has yet to be fully elucidated. Understanding the genetic basis for leaf morphology and biomass yield in sorghum facilitates the identification of key genes and genetic markers [13], enabling the application of molecular breeding techniques to introduce these traits into new crop varieties. These achievements can significantly enhance the precision and effectiveness of sorghum breeding efforts, ultimately contributing to global sustainable agriculture [6]. Thus, deciphering the genetic basis of leaf morphology and biomass yield is an essential component of breeding programs.

Leaf morphology regulation is controlled by both environmental factors and genetic regulators [14]. Several genes and pathways have been reported to be involved in leaf morphology regulation in many plants [15], and mutations in these genes frequently result in atypical leaf morphology in model plants. For example, a change in the *SERRATE* gene (SE) would lead to abnormal leaf morphology in Arabidopsis. The *SERRATE* gene encodes a zinc finger protein in Arabidopsis, and the mutant displays serrated leaves [16]. The *LATE MERISTEM IDENTITY1* (*LMI1*, AT5G03790) gene acts as a meristem identity regulator, which is essential for serrated leaves in Arabidopsis [17]. In cereal crops, NARROW LEAF 1 and NARROW LEAF 7 regulate the leaf morphology of rice via the auxin-mediated acid growth mechanism [12,18]. *GLYCINE MAX DWARF CRINKLED LEAF 1* (*GmDCL1*) is a promising gene involved in the morphogenesis of the crinkled leaf trait of the soybean [10]. These studies demonstrate that leaf morphology is a complex trait involved in many genetic pathways such as hormone signals, transcription factors and the interactions between them [15]. In sorghum, *BY1* influences both biomass and grain yield. This influence is exerted through the regulation of the primary and secondary metabolism, particularly via the shikimate pathway [11]. Though some studies reported the regulation of leaf morphology development, in sorghum, this trait is in need of further investigation.

The MutMap technique combines whole-genome sequencing and genetic analysis to facilitate gene mapping [19]. It can be used to rapidly identify molecular markers tightly linked to causal genes underlying a given phenotype [20]. MutMap employs high-generation segregating populations as mapping populations, thereby circumventing the necessity for backcrossing between mutant and wild-type parents. Consequently, it provides notable advantages in terms of rapidity and precision [21]. Many candidate genes associated with desirable agronomic traits have been isolated using the MutMap method [22]. For example, many rice grain-size-related genes such as *OML4* (Mei2-like protein 4), *OSH15* (*Oryza sativa* homeobox 15) and *PPKL1* (protein phosphatase with kelch-like domains 1) were isolated using MutMap method [23,24,25]. While numerous genes in rice have been mapped, the utilization of MutMap for identifying genes in sorghum mutants has not been documented.

In this study, we established an F2 inbred segregating line of a sorghum lobed-leaf mutant (*sblob*) which comprises a significant number of individuals manifesting the lobed-leaf phenotype, as well as individuals displaying the wild-type leaf phenotype. Then, NGS (next-generation sequencing)-based genotyping was performed for the mutant phenotype pool and the wild-type phenotype pool to identify the genome difference between the two pools, and 2M SNP markers were developed. Subsequently, MutMap analysis was conducted by employing the developed SNP markers and 9M candidate intervals, and 937 candidate genes were obtained. Finally, based on the functional annotation information of the developed SNPs and an analysis of the sequence differences between the two allelic pools, the candidate gene underlying the lobed leaf in sorghum was identified and predicted, and was designated as *sblob1*. The primary objective of this study is to identify the specific genetic locations and candidate genes associated with the development of lobed leaves in sorghum. In our work, *sblob1* (Sobic.003G010300) is suggested as a strong candidate gene accounting for the regulation of the lobed-leaf phenotype observed in the *sblob* mutant. This study also provides valuable insights into the regulatory processes accounting for lobed leaf formation in sorghum, thus supporting subsequent investigations in the field of sorghum functional genomics.

## 2. Results

### 2.1. Phenotypic and Physiology Analysis of sblob and WT Plants

Phenotypic analysis showed that the *sblob* and WT plants exhibited significant differences in plant morphology at the mature stage (Figure 1A), and the leaves of sblob plants exhibited a lobed phenotype at the heading stage (Figure 1B). Moreover, the leaf length, leaf width, leaf angle, panicle length, stem diameter, grain yield per panicle (Figure 1C) in the vegetative and mature stage (Figure 1D and Appendix A) of sblob plants decreased significantly, while differences in the plant height and hundred-grain weight between the two allelic pools showed no significance (Figure 1D and Appendix A).

In the traits exhibiting differential characteristics, the mutant displayed reductions when compared to the corresponding traits in the wild type. For example, the mutant displayed more than a 42% reduction in stem diameter and leaf angle and a 37% decrease in leaf length (Figure 1D and Appendix A). Nevertheless, it is worth highlighting that the sblob plants showed a decrease in leaf angle, indicating a compact plant architecture (Figure 1A).

Upon microscopic examination of leaf anatomical characteristics in both WT and *sblob* plants, the *sblob* plants exhibited an obvious decrease in the volume of leaf epidermal cells (Figure 2A and Appendix A) and leaf thickness (Figure 2B and Appendix A) compared to the WT plants (Figure 2A and Appendix A). Furthermore, observable differences were noted in the vascular tissues of both WT and *sblob* plants (Figure 2A and Appendix A). The vascular bundles in the *sblob* mutant displayed underdeveloped characteristics compared to their well-developed counterparts in WT plants (Figure 2A), while the parenchyma cell layers surrounding the vascular bundles in *sblob* plants exhibited less development when contrasted with those in WT plants (Figure 2D).

The assessment of chlorophyll levels in *sblob* and WT leaves during the shooting and heading stages revealed a potential correlation between the lobed-leaf phenotype and reduced chlorophyll content (Figure 2C), leading to diminished photosynthetic efficiency. Notably, at the heading stage, chlorophyll a levels in *sblob* plants decreased by over 40% compared to those of WT plants (Appendix A), concomitant with a nearly 20% reduction in photosynthetic efficiency (Figure 2E and Appendix A).

### 2.2. Genetic Analysis of the sblob Mutant

To investigate the genetic regulation of the *sblob* mutant, we established an F2 population comprising 412 individual plants, which was generated from a cross between *sblob* and 1383-2 (wild-type cultivar) plants. F1 plants were then cultivated and allowed to self-pollinate to generate F2 seeds for subsequent analyses. All F2 plants, along with the two parental cultivars, were grown in the field for segregation analysis. At the shooting stage, the distinctive mutant leaf phenotype could be clearly distinguished from the typical leaf morphology of WT plants. The segregation analysis of *sblob* in the F2 population was carried out using the Chi-square analysis method, as previously described [26,27]. This analysis revealed a theoretically expected Mendelian ratio of 3:1 for *sblob* to WT phenotype (Table 1). The segregation pattern observed in the experiment implies that the *sblob* phenotype is controlled by a single dominant mutation [28].

### 2.3. Mapping of the sblob1 Gene by Whole-Genome Resequencing

A total of 190,778,008 and 177,743,206 clean reads were generated from WT and *sblob* libraries, respectively, after data filtering was aligned with the sorghum reference genome. Subsequently, 28,728,254,222 and 26,767,472,954 clean bases were obtained from the clean-reads pools (Table 2). The percentage of high-quality bases (Q30) for WT and *sblob* pools was over 94%, which was indicative of their good quality (Table 2). Moreover, in WT and *sblob* pools, the mapping ratios were 99.12% and 98.49%, respectively (Appendix A), while the proper mapping ratios (%) were 95.68 and 95.13 (Table 2) and the average depths for the two pools were 44.16 and 40.98, respectively (Table 2).

After SNP calling, 2,060,083 SNPs and 418,196 indels were generated from the WT pools, while the *sblob* pools provided 2,061,011 SNPs and 418,310 indels, respectively (Appendix A). The whole-genome MutMap analysis was performed using the ED algorithm. The results revealed that the total length of candidate regions was 9MB, comprising 22 intervals distributed on different chromosomes (Table 3 and Appendix A). Within these candidate intervals, 960 genes were annotated, accounting for 1328 transcripts (Table 3 and Appendix A).

### 2.4. Identification and Validation of Candidate Gene Accounting for sblob Mutant

MutMap analysis sequencing was conducted on distinct pools of recessive and dominant individuals, following the approach outlined previously [29]. The results allowed us to pinpoint the candidate gene within specific regions in three chromosomes (Figure 3A). Subsequently, the candidate SNPs in the pool of recessive individuals were analyzed and were found to exhibit heterozygosity in the dominant individual pool. Only one SNP met the specified criterion within the mapped interval. This particular SNP, located in the second exon of the Sobic.003G010300 gene, manifested a G-to-A alteration at position 885631bp, resulting in a singular amino acid change (G217S) (Figure 3B). Moreover, DNA sequence analysis confirmed the mutation site in the *sblob* mutant, and multiple protein sequence comparison indicated that the protein site was highly conserved in different crop plants (Figure 3C,D).

Protein BLAST analysis and Pfam analysis showed that Sobic.003G010300 encodes an SbCYP71A1 protein, which belongs to the Cytochrome P450 superfamily (CYP) (Figure 4A). Homology alignment and gene annotation revealed that the CYP71A1 gene codes for the tryptamine 5-hydroxylase enzyme (Figure 4B), a crucial component in the plant serotonin/melatonin biosynthesis pathway [30,31]. This confirmed that Sobic.003G010300 is a potential candidate gene for *sblob1*.

### 2.5. Structure Prediction and Molecular Docking

Given that the mutation site is located within the conserved domain of the sblob1 protein, and considering the distinct physical and chemical properties of Gly and Ser, this mutation could induce a structural alteration in the sblob1 protein. To verify this, Alphfold2 was employed to analyze the 3D structure. A substantial alteration was observed when the 217nd amino acid was switched from Gly to Ser (Figure 5A,B).

Furthermore, amino acids’ alteration of the conserved region of encoding enzymes generally causes structural variations, altering the binding models of the substrates. In this study, the substrate used for the sblob1 protein was tryptamine. In light of this, AutoDock 4.2.6 software was employed to perform molecular docking validations to examine the docking models of the sblob1 protein and its corresponding wild-type protein with the substrate. The results showed that the binding models of sblob1 with tryptamine changed significantly compared to the WT protein (Figure 5C,D). The WT protein interacted with tryptamine via the residues at the positions of 213(E) and 314(D), while the sblob interacted with the substrate at 489(A) and 492(L) (Figure 5C,D).

### 2.6. Determination of the Melatonin and Auxin Content in Both the WT Plant and sblob Mutant

For a more in-depth analysis of the physiology profile of the *sblob* mutant, the contents of melatonin and tryptamine in both *sblob* and WT plants were determined using LC-MS. No melatonin was detected in the *sblob* mutant plants, whereas a significant amount of melatonin was detected in the WT plants (Figure 6A and Appendix A). Comparatively, the tryptamine content in the *sblob* plants was significantly higher than that in the WT plants (Figure 6B and Appendix A). All types of auxins were measured, and their content displayed no significant difference between the WT and *sblob* plants (Figure 6C).

## 3. Discussion

### 3.1. Abnormal Leaf Phenotype and Cell Development Are Responsible for sblob Mutant

Biomass yield and leaf morphology are crucial agronomic traits in sorghum, and are key to determining crop productivity and overall plant performance [32]. Understanding the underlying molecular mechanisms that regulate these traits is essential for crop improvement and breeding programs.

The biomass yield of sorghum refers to the amount of above-ground plant material produced, which directly influences forage production [33]. It is directly influenced by leaf morphology, another important trait in sorghum [34] that affects photosynthetic efficiency, water use efficiency, and overall plant growth and development [35]. Although biomass yield and leaf morphology are key agronomic traits in sorghum, their molecular mechanisms remain poorly understood. Deciphering these mechanisms is crucial for enhancing crop productivity and developing improved sorghum varieties.

The biomass of the *sblob* mutant exhibited a reduction compared to the wild-type (WT) plant. This change could be attributed to various factors, including morphological abnormalities such as diminished stem diameter and shorter leaves (Figure 1). This compromised development at the top of the panicle and decreased the panicle length. Additionally, a lower seed-setting rate in *sblob* plants contributed to a diminished grain yield per plant (Figure 1). Microscopic examinations involving paraffin sectioning implied that the reduction in stem diameter and shorter and thinner leaves in *sblob* plants was a result of inhibited cell development (Figure 2A,B), which affects the chlorophyll content and photosynthesis efficiency (Figure 2C–E). These cellular- and physiological-level alterations ultimately led to the observed reduction in biomass.

### 3.2. Detection of Candidate Gene Accounting for sblob by MutMap

In recent research, the whole-genome sequencing (WGS) and MutMap approach has emerged as a prominent method in gene mapping studies across various plant species [36,37]. This technique could generate comprehensive data, covering extensive genomic regions on the reference genome with heightened sequencing depth. In this study, employing WGS on the two sample bulks derived from the F2 population yielded 56 GB of clean bases, with a Q30 over 94%, indicating the high quality of the bases (Table 2). Moreover, in WT and *sblob* pools, the mapping ratios were 99.12% and 98.49%, respectively (Table 2, Appendix A), and the average depths for the two pools were 44.16 and 40.98 (Table 2). These results are indicative of the accuracy and high quality of the sequencing library, and it was well suited to achieving precise detection of the candidate genomic region and genes associated with the target lobed-leaf phenotype.

The SNPs were precisely positioned across the complete array of sorghum chromosomes (at least one base coverage), resulting in a more concentrated genome coverage and facilitating an optimal marker density for the identification of the pertinent candidate gene. Subsequently, the candidate region for the SNPs was analyzed, with the aim of pinpointing the most likely gene associated with a given SNP and lobed-leaf trait.

### 3.3. Single Nucleotide Mutations Result in Gene Functional Inactivation of sblob1

In this study, the candidate SNP was isolated using MutMap, and the sequence differences between the two allelic pools were based on the previous study [38], which demonstrated this SNP’s efficiency and accuracy in other crops such as rice (*Oryza sativa*) [28], soybean (*Glycine max*) [10] and *Setaria italic* [39]. In our study, sequencing verification and homologous comparison results showed that a Gly to Ser amino acid substitution in a highly conserved region of Sobic.003G010300 (Figure 3) was responsible for the low biomass yield and lobed-leaf phenotype of the *sblob* mutant. Pfam and homology analysis identified the candidate gene as *SbCYP71A1* (Figure 4A), which was involved in the synthesis of melatonin/serotonin in plants (Figure 4B) [30]. Gly to Ser amino acid substitution in the candidate gene led to abnormal protein structure and functional loss (Figure 5); the melatonin level was reduced in the *sblob* mutant (Figure 6A), causing further phenotype abnormalities and decreased biomass yield.

In this study, MutMap and homolog-guided alignment were used to identify the candidate gene responsible for abnormal leaf morphology in sorghum. Compared with conventional map-based cloning, MutMap can rapidly identify candidate regions using F2 progeny derived from a cross between the mutant and its wild-type counterpart [40]. When coupled with homologous gene alignment and comprehensive gene annotation, this approach allows a more rapid screening of candidate genes [41]. This strategy has been successfully applied to the screening of candidate genes associated with high production and responses to both biotic and abiotic stress across various plant species [39,42]. Thus, we have further substantiated the reliability of MutMap and homolog-guided gene discovery in sorghum, extending the application of this methodology within the sorghum context. This expansion serves as a valuable reference for the investigation of sorghum mutants in future studies.

### 3.4. Sblob1 Regulates Sorghum Biomass by Influencing Melatonin Synthesis

Melatonin is involved in various aspects of the growth and development of plants, such as flowering, photosynthesis, growth rhythms, leaf and root morphogenesis, seed germination and other growth and development processes [43,44]. For example, in cereal crops, transgenic rice expressing higher levels of endogenous melatonin demonstrated increased biomass yield compared to wild-type plants [45], while in horticultural plants, the application of exogenous melatonin enhances biomass yield [46], total carbohydrates and nutritional value [47]. In bioenergy crops, such as switchgrass (*Panicum virgatum*), transgenic plants with elevated melatonin levels exhibited significantly improved growth characteristics and a higher biomass yield [48]. Conversely, a reduction in melatonin levels induced abnormal leaf morphology and reduced biomass and grain yield in rice, which was consistent with our findings [49].

Furthermore, several homologous genes of *sblob1* (*SbCYP71A1*) occur in various grass species; however, many remain largely unexplored. One notable exception is *OsCYP71A1* in rice. *OsCYP71A1* is associated with melatonin synthesis, encoding a protein with analogous functionality in sorghum [50]. There is also evidence that *OsCYP71A1* may be a promising target for enhancing rice production under adverse environmental conditions [51]. In this study, the *sblob* mutant also exhibited abnormal melatonin levels and reduced biomass yield, suggesting that *sblob1* is a candidate gene that regulates sorghum biomass by influencing melatonin synthesis.

Although there was a difference in tryptamine content between *sblob* and wild-type plants (Figure 6B), the differences in total auxin content showed no significance (Figure 6C), proving that the mutant phenotype was caused by the difference in melatonin content caused by variations of *sblob1*. However, no similar studies and genes have been conducted for sorghum. Therefore, *sblob1* has been identified as a novel gene that regulates the biomass and leaf morphology of sorghum.

Overall, our findings have identified *sblob1* as a candidate gene that regulates the biomass yield of sorghum. These findings have significant implications for the development of crops with greater biomass yields. Further research is needed to explore the molecular mechanisms by which melatonin regulates biomass yield and leaf morphology and to explore its potential as a target for genetic engineering in the context of crop improvement.

## 4. Materials and Methods

### 4.1. Construction of Mapping Population

The lobed-leaf mutant *sblob* was isolated after multiple generations from the ethyl–methane sulfonate (EMS)-treated wild-type cultivar 1383-2. The seeds of the wild-type cultivar “1383-2” were chosen for mutagenesis based on the findings of a previous study [52] and subsequently sown to establish the mutant population, yielding more than three hundred thousand (300,000) plants in the year of 2021. The *sblob* mutant was identified through careful phenotypic analysis, in which a systematic and comprehensive screening process was used to identify mutants that exhibited the intended phenotype. WT (wild-type) and *sblob* (mutant-type) plants were used for phenotypic and genetic analyses in this study. For the WT line and *sblob* line, all the plants were grown in the experimental station of Shanxi Agriculture University, Taigu, China.

The *sblob* mutants were easily distinguishable in the shooting stage, displaying a distinctive lobed-leaf phenotype. In recent years, more than 100 independent *sblob* mutants have been identified within this mutant population based on the previous study [53]. These putative *sblob* mutants were subsequently crossed with the WT (wild-type) line. The genetic nature of the mutation (recessive or dominant) was determined in the resulting F1 plants. If the F1 plants displayed a normal leaf phenotype, the mutation was characterized as recessive. On the contrary, if the F1 plants exhibited the *sblob* phenotype, the mutation was classified as dominant. Notably, all the *sblob* mutants identified have proven to be dominant. Only those *sblob* mutants that displayed a consistent Mendelian segregation ratio (3 *sblob*: 1 WT) in the F2 progeny were confirmed as genuine *sblob* mutants. Then, those F2 progeny lines were chosen as the mapping population with which to generate *sblob* and WT pools.

### 4.2. Field Experiment and Phenotype Analysis

The field experiments were conducted with three planting locations as replications. Each replication consisted of 4-row plots of 2 m in length. The row spacing and plant space were 60 and 6 cm, respectively.

During the vegetative growth stage, in which the segregating phenotypes could be observed in the F2 population, the individual plants were labeled based on the segregating phenotypes. Plant phenotypes such as plant height, flag leaf length, leaf angle, leaf width and panicle length were determined and measured in the flowering stage in the field (Appendix A), while the grain weight per plant and 100-grain weight were determined during the harvest stage.

The photosynthesis efficiency of WT and *sblob* plants was measured using the Li-6400 portable photosynthesis system (LI-COR, Lincoln, NE, USA) at the shooting and heading stage, with three biological replicates tested for each sample. Photosynthetic determination was conducted using the flag leaf from 08:00 to 11:00 on clear days. Each experiment involved three individual plants. To assess the chlorophyll content, fresh flag leaves from *sblob* and WT plants at the shooting and heading stages were harvested, weighed and ground to a powder in liquid nitrogen for further analysis. The chlorophyll content was extracted using acetone and measured according to a recently reported method [54].

### 4.3. Histological Analysis

For histological examination, fresh flag leaf tissues were harvested from both *sblob* and WT plants during the heading stage. The collected tissues were then immersed and fixed in a 3.7% FAA solution containing 3.7% formaldehyde, 70% ethanol and 5% glacial acetic acid. After overnight incubation at 4 °C, the specimens underwent a series of dehydration and infiltration processes. These samples were subsequently embedded in paraffin, cut into 8 µm slices using a Leica RM2265 microtome, stained with 1% Methylene Blue and observed using a BX51 microscope (Olympus, Tokyo, Japan).

### 4.4. DNA Extraction and Whole-Genome Sequence

Distinctive leaf phenotypes within the F2 population were chosen, and two genomic DNA bulks (WT and *sblob*) were established. Genomic DNA extraction was performed on fresh young leaves from F2 individuals using the CTAB extraction kit. Each bulk consisted of genomic DNA samples from 30 individual plants with equivalent concentrations. The concentration and quality of genomic DNA in each bulk were evaluated using a NanoDrop spectrophotometer (NanoDrop, Wilmington, DE, USA) and 1.2% agarose gel electrophoresis, respectively. After that, the DNA bulks were employed for library construction and subsequent whole-genome sequencing.

The genome libraries were constructed and verified based on the previous study [10]. Whole-genome sequencing was performed using the High Throughput Illumina Library Sequencing (Illumina HiSeq X ten) (Illumina, San Diego, California, USA) Platform. The raw data were filtered by the phred score to remove adapter sequences and low-quality bases to obtain the clean data, based on the previous study [55]. The clean data were then aligned and mapped to the reference genome of sorghum using BWA-MEME2 software (accessed on 8 August 2021) [56].

As the clean data were aligned on the reference genome, the mapped ratio, sequencing depth and genome coverage were calculated and evaluated. The average alignment efficiency of the reads was evaluated based on the Q30 score (the percentage of bases with a mass value of 30 or greater), which exceeded 80%, indicating the high quality of the bases.

### 4.5. SNP Calling and Annotation

SNP calling was performed using GATK’s HaplotypeCaller method, and the filtering criteria were applied according to the recommended parameters. Subsequently, the SnpEff program was utilized to perform functional annotation of identified SNPs and indels based on the genome annotation of sorghum [57]. Finally, the detailed position and functional annotations of each SNP and indel were obtained.

### 4.6. Association Analysis Based on Euclidean Distance (ED)

To select the candidate genome region associated with the phenotype of the *sblob* mutant, Euclidean distance (ED) analysis was performed based on the previous study [58]. In the present study, power processing was adopted to eliminate background noise [58]. Candidate regions were detected by using 99.5% as the threshold to avoid deviation caused by uneven distribution of SNP markers. It was determined that the candidate regions must contain at least 10 variant sites above the threshold. Finally, according to the results of the ED analysis, the Manhattan map was drawn using the R program.

### 4.7. Candidate Mutation Site Sequence

DNA was extracted from fresh leaves of both the wild-type (WT) and *sblob* plants using the CTAB method. Fragments encompassing the mutation sites were then amplified via PCR for subsequent sequencing. The PCR primers were designed and employed based on the sequence of 400 bp upstream and downstream of the mutation site, resulting in an amplification fragment of 800 bp to ensure the accuracy of sequencing. All the PCR primers are listed in Appendix A. Subsequently, the PCR fragment was constructed using the PMD-18T vector and sequenced by Sangon Biotech Ltd. (Sangon, Shanghai, China).

### 4.8. Bioinformatics Analysis and Function Prediction of Candidate Genes

A BLAST search was performed using the sblob1 protein sequence as a query in both the Phytozome (https://phytozome.jgi.doe.gov/pz/portal.html, accessed on 1 January 2023) and NCBI (https://www.ncbi.nlm.nih.gov, accessed on 1 January 2023) databases. An E-value threshold of 0.0001 was applied to identify sblob1 proteins across various species. All sblob1 protein sequences were aligned using CLUSTALX 2.1 (http://www.clustal.org/). The phylogenetic tree was constructed using MEGA 7.0 software in addition to the neighbor-joining method.

### 4.9. 3D Structural Prediction and Molecular Docking

In this study, the 3D structural characterization of two distinct proteins, the mutant protein (sblob1) and its wild-type counterpart (WT), was conducted using the AlphaFold algorithm (https://alphafold.com/, accessed on 15 March 2023) [59].

Then, the PDB files with the highest confidence level exported by AlphaFold were imported into AutoDockTools 1.5.6 software for molecular docking validation. The binding modes between the sblob1 protein and its wild-type counterpart (WT) protein with the substrate (tryptamine) were validated separately. Each docking validation was repeated at least twice. The 3D interaction model was then visualized by PyMOL (http://www.pymol.org/, accessed on 9 August 2022).

### 4.10. Detection of Tryptamine, Melatonin and Various Hormone Contents

The levels of tryptamine, melatonin and various kinds of plant auxin in both plant types were measured to further elucidate the physiological differences between the mutant plants and their corresponding wild-type counterparts. The flag leaves at the heading stage were selected and harvested as the primary material for hormone analysis. For the purpose of accuracy, the sample for each biological replicate contained at least five flag leaves of individual plants. Each experiment contained at least two biological replicates. The detection of tryptamine, melatonin and auxin was conducted at MetWare (http://www.metware.cn/, accessed on 5 May 2021) based on the AB Sciex QTRAP 6500 LC-MS/MS platform, which ensures accurate and reliable measurements [60]. The components were extracted according to the method described in a previous study [61]. The details of various kinds of auxin are listed in Appendix A.

## 5. Conclusions

In this study, we successfully identified and characterized the *sblob* mutant in sorghum. The *sblob1* gene underlying the *sblob* mutant was mapped using MutMap, and its function was predicted and investigated. The encoded protein, tryptamine 5-hydroxylase enzyme (SbCYP71A1), belongs to the Cytochrome P450 superfamily. Through bioinformatics analysis and physiology profiling, we gained insights into the role of *sblob1* in regulating sorghum biomass yield by influencing the melatonin synthesis pathway. These findings contribute to an understanding of the molecular mechanisms underlying sorghum growth and productivity, with potential implications for crop improvement.

## Figures and Tables

**Figure 1 ijms-25-00796-f001:**
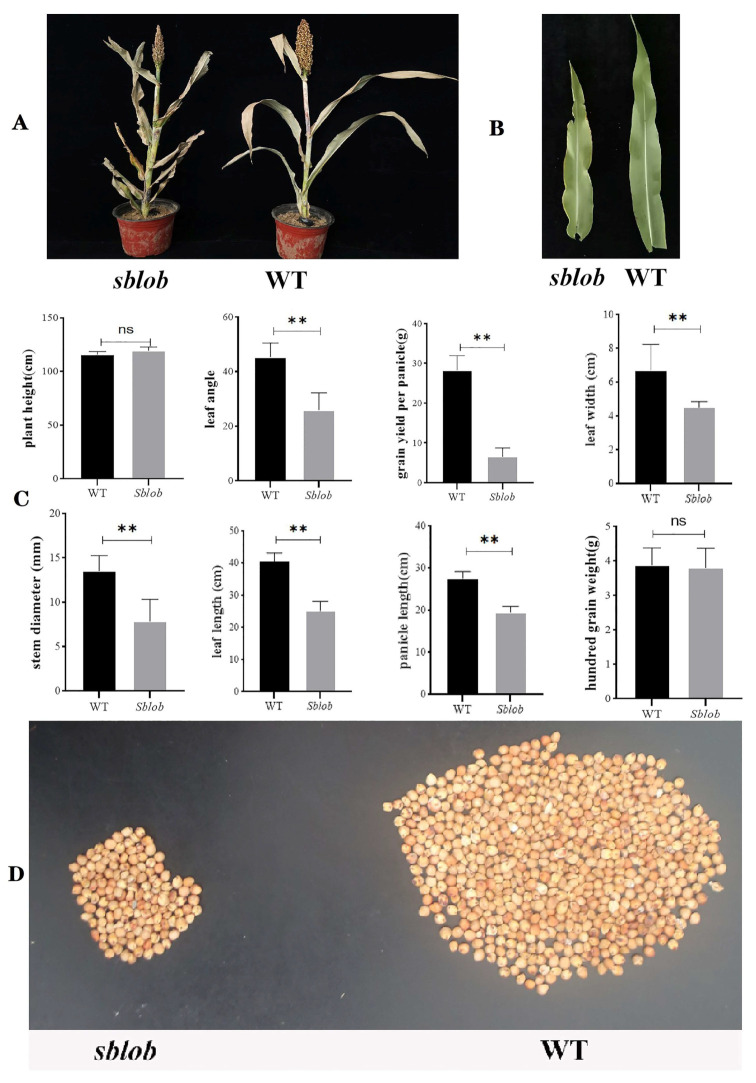
Assessment of phenotypic characteristics and agronomic traits of WT and *sblob* plants. (**A**) The phenotypes of WT and *sblob* at the harvest stage. (**B**) Flag leaf phenotypes of WT and *sblob* at the heading stage. (**C**) The grain yield per panicle and grain phenotype of WT and *sblob.* (**D**) Presentation of comparative analysis of differences in plant height, panicle length, flag leaf length, flag leaf width, grain weight per panicle, and 100-grain weight between WT and *sblob*. All values are represented as means ± SD (n = 3). ** Indicates a significant difference between groups of 0.01, while ns indicates no significant difference.

**Figure 2 ijms-25-00796-f002:**
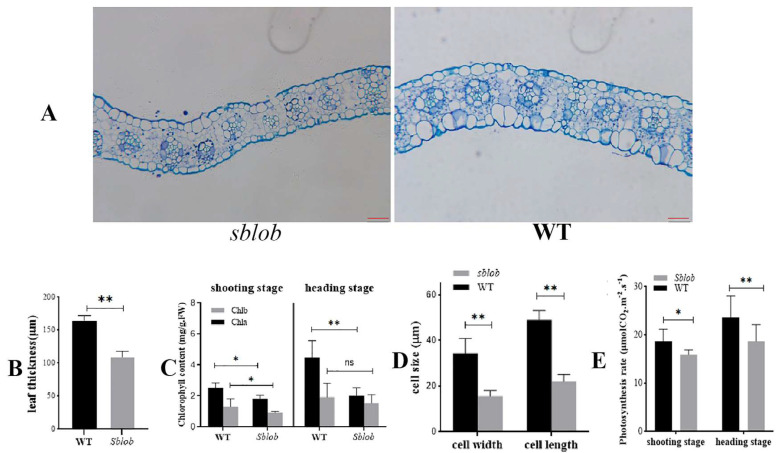
Microscopic examination of leaf anatomical characteristics and photosynthetic efficiency analysis of both WT and *sblob*. (**A**) Paraffin section observation of a leaf from WT and one from *sblob* at the shooting stage. The scale bar represents 50 μm. (**B**) Leaf thickness of WT and *sblob* at the shooting stage. (**C**) Chlorophyll content of *sblob* and WT plants at the shooting and heading stages. (**D**) Cell size analysis of leaf from WT and *sblob* at the shooting stage. (**E**) Photosynthesis efficiency analyses of WT and *sblob* at the shooting and heading stages. * And ** indicate significant differences between groups of 0.05 and 0.01, respectively, while ns indicates no significant difference.

**Figure 3 ijms-25-00796-f003:**
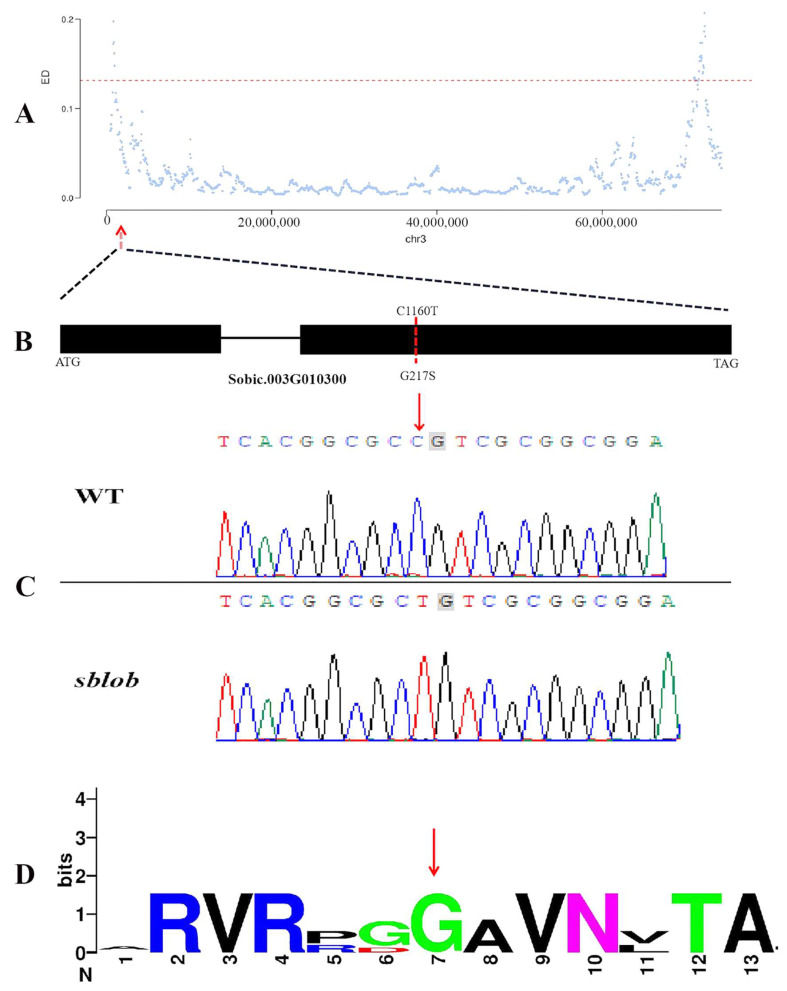
Mapping of *sblob1* gene. (**A**) Presentation of Euclidean distance (ED) correlation values across sorghum chromosome 3. Each blue dot represents the ED value of an individual single-nucleotide polymorphism (SNP) locus, while the red dashed line represents the significant correlation threshold. The red arrow indicates the identified candidate SNP locus. (**B**) Illustration of *sblob1* structure and its mutation site. The ATG and TGA were at the left and right ends, respectively, representing the start and stop codons. Exons are depicted by black boxes; introns are represented by lines. The red line and text in the second exon specify the position and nature of the base mutation in the *sblob* mutant. (**C**) The DNA sequence at the candidate mutant site in *sblob* and WT plants. The red arrow highlights the mutant site. (**D**) Analysis of conservation in the amino acid substitution region and its frequency among homologous plant genes. The red arrow indicates the position of the amino acid substitution in the *sblob* mutant.

**Figure 4 ijms-25-00796-f004:**
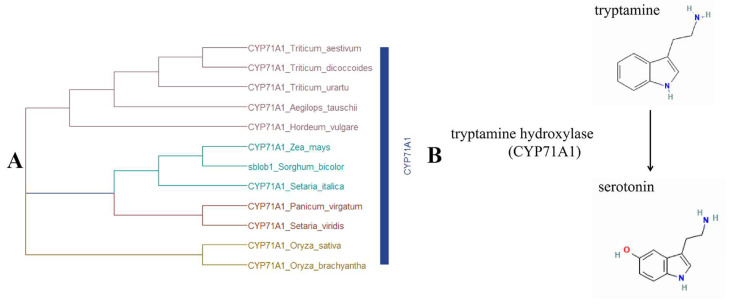
Phylogenetic analysis of and physiological pathways involved in *sblob*1. (**A**) Phylogenetic analysis of the candidate gene and its homologs established through protein sequencing. The construction of the phylogenetic tree employed the neighbor-joining (NJ) algorithm by using MEGA 7.0 Software. (**B**) *sblob1* is involved in the biological process of serotonin/melatonin synthesis.

**Figure 5 ijms-25-00796-f005:**
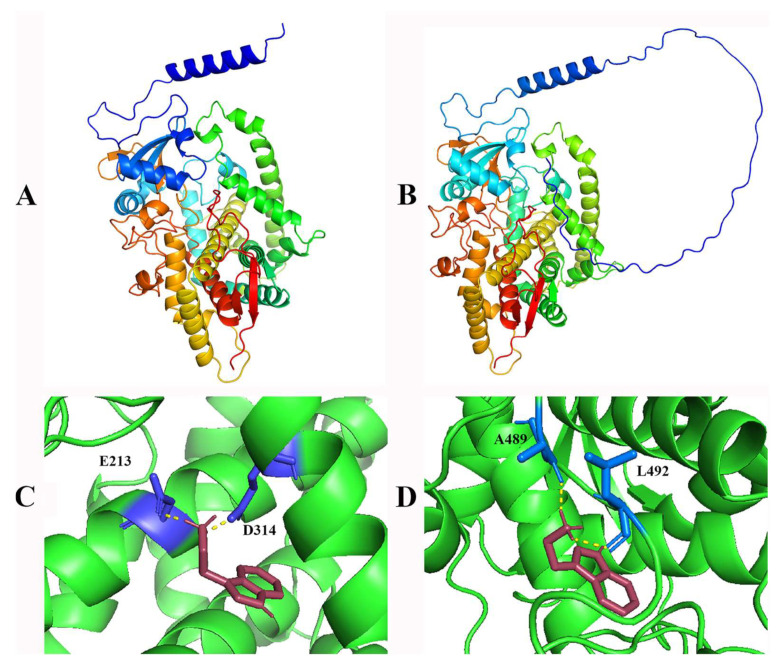
Three-dimensional structure analysis and molecular docking analysis of sblob1 protein and its WT counterpart. (**A**) Three-dimensional structure of WT (SbCYP71A1) protein. (**B**) Three-dimensional structure of sblob1 protein. (**C**) The binding pattern of WT (SbCYP71A1) protein and tryptamine. (**D**) The binding pattern of sblob1 protein and tryptamine. The brownish-red molecule is tryptamine.

**Figure 6 ijms-25-00796-f006:**
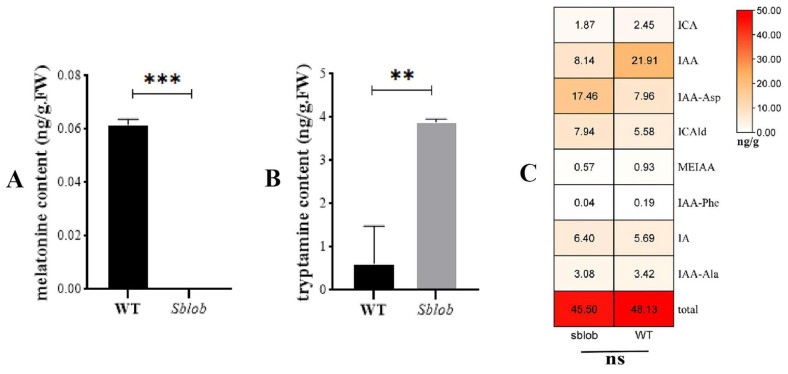
Melatonin, tryptamine and hormone content determination. (**A**) Melatonin content of *sblob* and WT. (**B**) Tryptamine content of *sblob* and WT. (**C**) Auxin content of *sblob* and WT. ** and *** indicate a significant difference between groups of 0.01 and 0.001, respectively, while ns indicates no significant difference.

**Table 1 ijms-25-00796-t001:** Chi-square analysis of trait segregation in the F2 population.

	WT	sblob	Total	χ^2^ (df = 1)
actual	118	294	412	
expected	103	309	412	2.63

**Table 2 ijms-25-00796-t002:** Summary of the sequencing data.

	Clean Reads	Clean Bases (bp)	Clean Q30 (%)	Mapped Ratio (%)	Proper Ratio (%)	Average Depths
WT	190,778,008	28,728,254,222	94.49	99.12	95.68	44.16
*sblob*	177,743,206	26,767,472,954	94.75	98.49	95.13	40.98

**Table 3 ijms-25-00796-t003:** The physical positions and numbers of genes in the 9 identified candidate genomic regions.

Region	Start Position (Mb)	End Position (Mb)	Region Size	Transcripts	Gene Number
chr10	57.91461	58.149969	0.235359	30	24
chr1	1.154111	1.34952	0.195409	46	39
chr1	1.409762	1.532592	0.12283	49	30
chr1	1.623681	1.691337	0.067656	15	14
chr1	1.812801	1.938655	0.125854	38	31
chr1	4.068754	4.498218	0.429464	63	52
chr1	58.306682	59.217158	0.910476	87	65
chr1	65.020891	65.164125	0.143234	16	15
chr1	6.504078	6.639419	0.135341	24	16
chr1	67.295568	67.722791	0.427223	57	49
chr1	72.061557	72.655387	0.59383	85	67
chr1	8.567085	8.898033	0.330948	63	29
chr2	4.668026	5.149803	0.481777	76	59
chr2	5.400013	5.487665	0.087652	7	5
chr2	5.664728	5.793508	0.12878	21	11
chr2	71.663212	72.648182	0.98497	190	116
chr3	0.703708	1.323768	0.62006	104	85
chr3	70.882922	72.678683	1.795761	285	199
chr4	60.551245	60.866205	0.31496	45	29
chr6	23.890052	24.617971	0.727919	0	0
chr6	59.233376	59.338064	0.104688	22	20
chr8	58.306666	58.339277	0.032611	5	5
total			8.996802	1328	960

## Data Availability

Data is contained within the article and Appendix A.

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
