# Peer review of "Mapping and Screening of Candidate Gene Regulating the Biomass Yield of Sorghum (Sorghum bicolor L.)"

_ijms, 2024, doi:10.3390/ijms25020796_

Round 1

Reviewer 1 Report

Comments and Suggestions for Authors

Here are my comments: 

• The introduction could be strengthened by providing more background on the importance of biomass yield and leaf morphology in sorghum specifically, and how understanding the genetic mechanisms regulating these traits can inform crop breeding efforts.

• Additional details are needed on the mutagenesis approach used to generate the sblob mutant population. What mutagen was used? What was the scale of the mutant population screened? This provides helpful context. 

• In the results section, the phenotypic differences observed between sblob mutant and wild-type plants should be quantified to the extent possible (e.g. percentage differences in traits like leaf length, biomass, etc.) to complement the visual representations. 

• The candidate gene sblob1 is predicted to encode a protein involved in melatonin biosynthesis based on homology. Directly measuring melatonin levels in sblob mutant vs. wild-type plants would strengthen the functional connection. 

• The model for sblob1's role regulating biomass and leaf morphology via melatonin could be better explained. Is there evidence that melatonin impacts these plant growth processes?

• For the mutmap analysis, additional details on the number of recessive vs dominant plants in the pools, sequencing depth, and parameters for candidate SNP filtering would provide helpful methodological context.

• The authors note sblob1 is a novel biomass/morphology regulatory gene in sorghum, but should comment on any homologs identified in other plant species and similarities/differences.

• The discussion could be expanded to place findings in a broader context about the utility of mutmap and homolog-guided gene discovery for identifying agronomically relevant loci.

Author Response

Dear Reviewer: 

We extend our sincere gratitude to the reviewer for their thoughtful and constructive feedback on our manuscript, ‘Mapping and Screening of Candidate Gene Regulating the Biomass Yield of Sorghum (Sorghum bicolor L.)’.To enhance clarity in our responses, we have adhered to a color-coded system: the original comments are presented in black font, our responses are in blue font, and any text added to the manuscript is highlighted in red font.

We greatly appreciate the reviewer's diligence and valuable insights, which have significantly contributed to the improvement of our work. Your commitment to ensuring the rigor and quality of the manuscript is sincerely acknowledged. Thank you for your continued support and guidance throughout the review process.

Reviewer 1: 

The introduction could be strengthened by providing more background on the importance of biomass yield and leaf morphology in sorghum specifically, and how understanding the genetic mechanisms regulating these traits can inform crop breeding efforts.

 Response: Thank you for forwarding the reviewer's valuable suggestion regarding the strengthening of the introduction in our manuscript. We appreciate the insightful feedback and are committed to enhancing the manuscript accordingly. The importance of the biomass and leaf morphology were added in the manuscript and marked in red.

The high biomass yield of sorghum translates into a substantial quantity of plant material, offering abundant feedstock for biofuel production[1]. For example, sweet-stemmed sorghum emerges as a valuable resource for extracting sugars[2], while the leaves and stems contribute to the production of cellulose—a key component in biofuel and bioethanol manufacturing processes [1-3]. Therefore, the biomass yield of sorghum is important for its potential in energy production, as well as for its economic viability and diverse agricultural applications[4].

The importance of leaf morphology was added as follows:

As a potential energy crop, leaf morphology is one of the key agronomic traits affecting the biomass yield of sorghum with a broad range of phenotypic variations[5]. For example, larger leaf areas have been found to be positively correlated with biomass yield characters in sorghum[6]. Moreover, leaf morphology can also affect nutrient uptake, which is crucial for the plant's growth and biomass yield[7].

The content on understanding the genetic mechanisms regulating these traits can inform crop breeding efforts were added as follows:

Although leaf morphology and biomass yield are important agronomic traits, the molecular genetic mechanism that controls these trait has not been clearly elucidated in sorghum. Understanding the genetic basis for leaf morphology and biomass yield in sorghum facilitates the identification of key genes and genetic markers[8]. This in turn enables the application of molecular breeding techniques to introduce these traits into new crop varieties. All of these achievements can significantly enhance the precision and effectiveness of sorghum breeding efforts, ultimately contributing to global sustainable agriculture[4]. Thus, it is essential to deciphering the genetic basis of leaf morphology and biomass yield in breeding program.

 This expanded section aims to provide a more comprehensive exploration of sorghum's significance in the realm of biomass yield and leaf morphology, illuminating its crucial role in both bioenergy production and sustainable agriculture. Thank you again for your suggestions.

  • Additional details are needed on the mutagenesis approach used to generate the sblob mutant population. What mutagen was used? What was the scale of the mutant population screened? This provides helpful context. 

 Response: we are grateful for the opportunity to provide additional details on the mutagenesis approach used to generate the sblob mutant population, as suggested.

Mutagenesis Approach:

The sblob mutant population was generated through ethyl methanesulfonate (EMS) mutagenesis. This mutagenesis approach was chosen for its ability to introduce heritable genetic variations in a targeted and controlled manner.

Scale of the Mutant Population:

The mutagenesis process involved the treatment of a population of three hundred thousand (300,000) plants. This large-scale population was subjected to EMS mutagenesis to increase the likelihood of capturing a diverse array of mutations, ensuring a comprehensive pool for subsequent screening.

Screening Process:

The sblob mutant was identified through careful phenotypic analysis. We employed a systematic and thorough screening process to select mutants exhibiting the desired phenotype. We have add these content in the material and method part which was highlighted in red.

The sblob mutant population was created using ethyl methanesulfonate (EMS) mutagenesis. The seeds of the wild-type cultivar "1383-2" were chosen for mutagenesis based on a prior study[9]. Subsequently, the seeds were sown to establish the mutant population, yielding more than three hundred thousand (300,000) plants in the year of 2021. The sblob mutant was identified through carefully phenotypic analysis, employing a systematic and comprehensive screening process to identify mutants exhibiting the intended phenotype.

Thank you for the valuable suggestion, and we remain dedicated to enhancing the contextual information in our manuscript.

  • In the results section, the phenotypic differences observed between sblob mutant and wild-type plants should be quantified to the extent possible (e.g. percentage differences in traits like leaf length, biomass, etc.) to complement the visual representations. 

Response: thank you for your advice. To complement the visual representations, we have incorporated quantitative data on phenotypic differences in a dedicated table S1. This table provides percentage differences in key traits, including leaf length, biomass yield, and other relevant parameters, offering a detailed and quantitative perspective on the observed variations between the sblob mutant and wild-type plants. 

Table S1

traits

WT

sblob

reducthion rate(%)

plant height/cm

115.87±2.68

119±3.34 ns

ns

hundred grain weight/g

3.87±0.5

3.79±0.57 ns

ns

leaf angle

45.34±4.93

25.94±5.98 **

-42.8

stem diameter/mm

13.5±1.65

7.38±2.38**

-42

leaf length/cm

40.75±2.37

25.76±2.73**

-37.74

leaf width/cm

6.74±1.51

4.52±0.36**

-32.9

pannicle length/cm

27.69±1.54

19.55±1.42**

-29.43

grain yield per panicle/g

28.67±3.67

6.55±2.19**

-76.81

leaf thickness/um

162.46±7.53

108.79±9.02**

-33.03

cell width/um

39.09±7.32

17.76±2.74**

-54.5

cell length/um

55.75±4.59

24.64±2.87**

-55.8

chla at shooting stage/mg.g-1.FW

2.51±0.31

1.81±0.23*

-21.9

chla at heading stage/mg.g-1.FW

4.49±1.07

2.01±0.5**

-55.2

chlb at shooting stage/mg.g-1.FW

1.3±0.5

0.91±0.07*

-20.8

chlb at heading stage/mg.g-1.FW

1.92±0.89

1.52±0.55 ns

ns

Photosynthesis rate at shooting stage/umol CO2.m-2.s-1

18.6±2.58

15.9±1*

-14.7

Photosynthesis rate at heading stage/umol CO2.m-2.s-1

23.67±4.4

18.7±3.43**

-20.9

Data are presented as the mean±SD based on three individuals. Asterisks indicate a significant difference between WT and sblob: n=3, Welch’s two-sample t-test,

*P<0.05, **p<0.01, ns no significance

We believe that the inclusion of quantitative data in the results section significantly enhances the robustness of our findings. We appreciate the reviewer's suggestion, and we hope these additions contribute to the clarity and completeness of our manuscript.

  • The candidate gene sblob1 is predicted to encode a protein involved in melatonin biosynthesis based on homology. Directly measuring melatonin levels in sblob mutant vs. wild-type plants would strengthen the functional connection. 

 Response: We sincerely appreciate the reviewer's recognition and thoughtful feedback. We would like to clarify that our study indeed includes a robust approach to validating the functional connection of sblob1. In both the Materials and Methods section and Figure 6, as well as supplementary Figure S3 and S4, we have outlined the methodology for directly measuring melatonin levels in sblob mutant and wild-type plants.

We have taken the reviewer's suggestion into careful consideration, and we trust that the information provided in the specified sections of our manuscript addresses the functional connection through the direct measurement of melatonin levels.

Once again, we extend our gratitude to the reviewer for their valuable insights and for contributing to the improvement of our manuscript.

Thank you for your continued support.

  • The model for sblob1's role regulating biomass and leaf morphology via melatonin could be better explained. Is there evidence that melatonin impacts these plant growth processes?

Response: In response to the query regarding the model for sblob1's role in regulating biomass and leaf morphology via melatonin, we want to address the question raised by the reviewer and provide additional context in the discussion section 3.4 as follows:

Melatonin is deeply involved in the growth and development of plants, such as flowering, photosynthesis, growth rhythms, leaf and root morphogenesis, seed germination, and other growth and development processes [10,11].For example, in cereal crops, transgenic rice expressing higher levels of endogenous melatonin demonstrated increased biomass yield compared to wild-type plants[12]. In horticultural plants, the application of exogenous melatonin has been shown to enhance biomass yield[13], total carbohydrates and nutritional value [14]. In bioenergy crops, such as switchgrass (Panicum virgatum), transgenic plants with elevated melatonin levels exhibited significantly improved growth characteristics and higher biomass yield[15]. Conversely, a reduction in melatonin levels was found to induce abnormal leaf morphology, reduced biomass, and grain yield in rice, which was consistent with findings in our study [16].

We appreciate the reviewer's valuable suggestion, and we are committed to enhancing the clarity and scientific merit of our manuscript.

  • For the mutmap analysis, additional details on the number of recessive vs dominant plants in the pools, sequencing depth, and parameters for candidate SNP filtering would provide helpful methodological context.

Response: We would like to express our gratitude to the reviewer for their meticulous review and valuable comments. We have taken the reviewer's suggestions into consideration, and we would like to highlight that the number of recessive vs dominant plants in the pools, sequencing depth, and parameters for candidate SNP filtering are comprehensively detailed in the material and methods section and Table S3 and S4. We believe that the inclusion of these supplementary tables addresses the methodological context of our mutmap analysis, providing transparency and clarity for readers and fellow researchers. Once again, we appreciate the thorough review and constructive feedback provided by the reviewer.

Thank you for your continued support.

  • The authors note sblob1 is a novel biomass/morphology regulatory gene in sorghum, but should comment on any homologs identified in other plant species and similarities/differences.

Response: We acknowledge the importance of discussing homologs of sblob1 in other plant species and highlighting any similarities or differences. In the revised version, we included a dedicated section addressing this aspect, which was added in the section 3.4 as follows:

Furthermore, there are several homologous genes of sblob1 (SbCYP71A1) across various grass species; however, many of them remain largely unexplored except for OsCYP71A1 in rice. OsCYP71A1 has been established to be associated with melatonin synthesis, encoding a protein with analogous functionality in sorghum [17]. There is also evidence suggesting that OsCYP71A1 could be a promising target for enhancing rice production under adverse environmental conditions[18]. In this study, sblob mutant also exhibited abnormal melatonin levels and reduced biomass yield. It is, therefore, suggested that sblob1 is a candidate gene regulating sorghum biomass by influencing melatonin synthesis.

We appreciate the reviewer's insightful feedback and remain committed to incorporating these suggestions to enhance the manuscript.

  • The discussion could be expanded to place findings in a broader context about the utility of mutmap and homolog-guided gene discovery for identifying agronomically relevant loci.

Response: We acknowledge the reviewer's valuable input and agree that discussing the broader utility of MutMap and homolog-guided gene discovery is crucial. In our revised discussion section 3.3, this content was added as follows:

In the present study, mutmap and homolog-guided alignment were employed to identify the candidate gene responsible for abnormal leaf morphology in sorghum. In comparison with conventional map-based cloning, mutmap method can rapidly identify candidate regions by using F2 progeny derived from the cross of the mutant and its wild type counterpart[19]. When coupled with homologous gene alignment and comprehensive gene annotation, this approach enables a more rapid screening of candidate genes[20]. This strategy has demonstrated successful application in screening candidate genes associated with high production and responses to both biotic and abiotic stress across various plant species [21,22]. We have further substantiated the reliability of mutmap and homolog-guided gene discovery in sorghum, extending the application of this methodology within the sorghum context. This expansion serves as a valuable reference for the investigation of sorghum mutants in future studies.

We appreciate the reviewer's guidance and remain committed to delivering an improved manuscript.

References:

  1. Yang, K.-W.; Chapman, S.; Carpenter, N.; Hammer, G.; McLean, G.; Zheng, B.; Chen, Y.; Delp, E.; Masjedi, A.; Crawford, M.; et al. Integrating crop growth models with remote sensing for predicting biomass yield of sorghum. in silico Plants 2021, 3, doi:10.1093/insilicoplants/diab001.
  2. Chiluwal, A.; Bheemanahalli, R.; Perumal, R.; Asebedo, A.R.; Bashir, E.; Lamsal, A.; Sebela, D.; Shetty, N.J.; Krishna Jagadish, S.V. Integrated aerial and destructive phenotyping differentiates chilling stress tolerance during early seedling growth in sorghum. Field Crops Research 2018, 227, 1-10, doi:/10.1016/j.fcr.2018.07.011.
  3. Silva, T.N.; Thomas, J.B.; Dahlberg, J.; Rhee, S.Y.; Mortimer, J.C. Progress and challenges in sorghum biotechnology, a multipurpose feedstock for the bioeconomy. Journal of Experimental Botany 2021, 73, 646-664, doi:10.1093/jxb/erab450.
  4. Baye, W.; Xie, Q.; Xie, P. Genetic Architecture of Grain Yield-Related Traits in Sorghum and Maize. Int J Mol Sci 2022, 23, doi:10.3390/ijms23052405.
  5. Zhi, X.; Tao, Y.; Jordan, D.; Borrell, A.; Hunt, C.; Cruickshank, A.; Potgieter, A.; Wu, A.; Hammer, G.; George-Jaeggli, B.; et al. Genetic control of leaf angle in sorghum and its effect on light interception. Journal of Experimental Botany 2021, 73, 801-816, doi:10.1093/jxb/erab467.
  6. Chen, J.; Zhu, M.; Liu, R.; Zhang, M.; Lv, Y.; Liu, Y.; Xiao, X.; Yuan, J.; Cai, H. BIOMASS YIELD 1 regulates sorghum biomass and grain yield via the shikimate pathway. Journal of Experimental Botany 2020, 71, 5506-5520, doi:10.1093/jxb/eraa275.
  7. Wang, D.; Liu, H.; Li, K.; Li, S.; Tao, Y. Genetic analysis and gene mapping of a narrow leaf mutant in rice (Oryza sativa L.). Chinese Science Bulletin 2009, 54, 752-758, doi:10.1007/s11434-009-0098-2.
  8. Wang, T.; Crawford, M.M.; Tuinstra, M.R. A novel transfer learning framework for sorghum biomass prediction using UAV-based remote sensing data and genetic markers. Front Plant Sci 2023, 14, 1138479, doi:10.3389/fpls.2023.1138479.
  9. Jiao, Y.; Nigam, D.; Barry, K.; Daum, C.; Yoshinaga, Y.; Lipzen, A.; Khan, A.; Parasa, S.-P.; Wei, S.; Lu, Z.; et al. A large sequenced mutant library – valuable reverse genetic resource that covers 98% of sorghum genes. The Plant Journal n/a, doi:doi.org/10.1111/tpj.16582.
  10. Sun, C.; Liu, L.; Wang, L.; Li, B.; Jin, C.; Lin, X. Melatonin: A master regulator of plant development and stress responses. J Integr Plant Biol 2021, 63, 126-145, doi:10.1111/jipb.12993.
  11. Zhang, Z.; Zhang, Y. Melatonin in plants: what we know and what we don’t. Food Quality and Safety 2021, 5, doi:10.1093/fqsafe/fyab009.
  12. Byeon, Y.; Back, K. An increase in melatonin in transgenic rice causes pleiotropic phenotypes, including enhanced seedling growth, delayed flowering, and low grain yield. Journal of Pineal Research 2014, 56, 408-414, doi:doi.org/10.1111/jpi.12129.
  13. Heydarnajad Giglou, R.; Torabi Giglou, M.; Esmaeilpour, B.; Padash, A.; Ghahremanzadeh, S.; Sobhanizade, A.; Hatami, M. Exogenous melatonin differentially affects biomass, total carbohydrates, and essential oil production in peppermint upon simultaneous exposure to chitosan-coated Fe3O4 NPs. South African Journal of Botany 2023, 163, 135-144, doi:doi.org/10.1016/j.sajb.2023.10.038.
  14. Agathokleous, E.; Zhou, B.; Xu, J.; Ioannou, A.; Feng, Z.; Saitanis, C.J.; Frei, M.; Calabrese, E.J.; Fotopoulos, V. Exogenous application of melatonin to plants, algae, and harvested products to sustain agricultural productivity and enhance nutritional and nutraceutical value: A meta-analysis. Environmental Research 2021, 200, 111746, doi:doi.org/10.1016/j.envres.2021.111746.
  15. Huang, Y.-H.; Liu, S.-J.; Yuan, S.; Guan, C.; Tian, D.-Y.; Cui, X.; Zhang, Y.-W.; Yang, F.-Y. Overexpression of ovine AANAT and HIOMT genes in switchgrass leads to improved growth performance and salt-tolerance. Scientific Reports 2017, 7, 12212, doi:10.1038/s41598-017-12566-2.
  16. Ke, S.; Liu, S.; Luan, X.; Xie, X.M.; Hsieh, T.F.; Zhang, X.Q. Mutation in a putative glycosyltransferase-like gene causes programmed cell death and early leaf senescence in rice. Rice (N Y) 2019, 12, 7, doi:10.1186/s12284-019-0266-1.
  17. Lee, K.; Choi, G.H.; Back, K. Inhibition of Rice Serotonin N-Acetyltransferases by MG149 Decreased Melatonin Synthesis in Rice Seedlings. Biomolecules 2021, 11, doi:10.3390/biom11050658.
  18. Mannino, G.; Pernici, C.; Serio, G.; Gentile, C.; Bertea, C.M. Melatonin and Phytomelatonin: Chemistry, Biosynthesis, Metabolism, Distribution and Bioactivity in Plants and Animals—An Overview. International Journal of Molecular Sciences 2021, 22, 9996.
  19. Cao, Z.Z.; Lin, X.Y.; Yang, Y.J.; Guan, M.Y.; Xu, P.; Chen, M.X. Gene identification and transcriptome analysis of low cadmium accumulation rice mutant (lcd1) in response to cadmium stress using MutMap and RNA-seq. BMC Plant Biology 2019, 19, 250, doi:10.1186/s12870-019-1867-y.
  20. Jiao, Y.; Burow, G.; Gladman, N.; Acosta-Martinez, V.; Chen, J.; Burke, J.; Ware, D.; Xin, Z. Efficient Identification of Causal Mutations through Sequencing of Bulked F2 from Two Allelic Bloomless Mutants of Sorghum bicolor. Frontiers in Plant Science 2018, 8, doi:10.3389/fpls.2017.02267.
  21. Wang, H.; Tang, S.; Zhi, H.; Xing, L.; Zhang, H.; Tang, C.; Wang, E.; Zhao, M.; Jia, G.; Feng, B.; et al. The boron transporter SiBOR1 functions in cell wall integrity, cellular homeostasis, and panicle development in foxtail millet. The Crop Journal 2022, 10, 342-353, doi:doi.org/10.1016/j.cj.2021.05.002.
  22. Zhang, S.; Abdelghany, A.M.; Azam, M.; Qi, J.; Li, J.; Feng, Y.; Liu, Y.; Feng, H.; Ma, C.; Gebregziabher, B.S.; et al. Mining candidate genes underlying seed oil content using BSA-seq in soybean. Industrial Crops and Products 2023, 194, 116308, doi:doi.org/10.1016/j.indcrop.2023.116308.

Reviewer 2 Report

Comments and Suggestions for Authors

This interesting paper describes the identification of a candidate gene that causes altered leaf morphology and reduced yield in a lobed leaf mutant of sorghum. This gene encodes tryptamine 5-hydroxylase from the Cytochrome P450 superfamily, which plays an important role in the serotonin/melatonin biosynthesis pathway. This manuscript could be published after revisions.

L.53. The second “and” should be removed.

L.73. An explanation for the abbreviation NGS should be provided.

The two paragraphs (L.71-81 and L.82-89) should be combined into one and rewritten.

Was the experiment performed with potted plants (Fig. 1A) or in the field (subsection 4.2)?

L.123. “The assessment of chlorophyll levels...”. Analysis of chlorophyll content is not included in the Materials and Methods. Add please.

L. 213. “the CYP71A1 gene codes ... a crucial component in the plant serotonin/melatonin biosynthesis pathway.” Why was the content of melatonin analyzed, and not serotonin, the synthesis of which is catalyzed by the product of the CYP71A1 gene (Fig. 4B)?

L.243. “the content of melatonin and tryptamine...”. Tryptamine content analysis is not included in the Materials and Methods. Add please.

Fig. 1. The leaf width of WT and sblob plants are almost the same in Fig. 1B, but differs by 1.5 times in Fig. 1D. Clarify please.

Fig. 6. Tryptamine is not melatonin or a hormone. The title of the figure should be corrected.

Subsection 4.10. The hormones analyzed and the sampling and analysis methods should be specified.

Fig. S3, S4. Please indicate the differences between the right and left parts of the figures.

Fig. S4. The title of the figure is written as S3. Correct please.

Table S1. “uesd” should be corrected to “used”.

Author Response

Dear reviewer: 

We extend our sincere gratitude to the reviewer for their thoughtful and constructive feedback on our manuscript, ‘Mapping and Screening of Candidate Gene Regulating the Biomass Yield of Sorghum (Sorghum bicolor L.)’.To enhance clarity in our responses, we have adhered to a color-coded system: the original comments are presented in black font, our responses are in blue font, and any text added to the manuscript is highlighted in red font.

We greatly appreciate the reviewer's diligence and valuable insights, which have significantly contributed to the improvement of our work. Your commitment to ensuring the rigor and quality of the manuscript is sincerely acknowledged. Thank you for your continued support and guidance throughout the review process.

Reviewer 2:

This interesting paper describes the identification of a candidate gene that causes altered leaf morphology and reduced yield in a lobed leaf mutant of sorghum. This gene encodes tryptamine 5-hydroxylase from the Cytochrome P450 superfamily, which plays an important role in the serotonin/melatonin biosynthesis pathway. This manuscript could be published after revisions.

L.53. the second “and” should be removed.

Response: The second “and” has been removed as suggested. We apologize for our carelessness.

L.73. An explanation for the abbreviation NGS should be provided.

Response: L.73: We have added an explanation for the abbreviation NGS: "NGS refers to next-generation sequencing” which has been marked in red in the revised manuscript.

The two paragraphs (L.71-81 and L.82-89) should be combined into one and rewritten.

Response: We appreciate your suggestion to combine the two paragraphs (L.71-81 and L.82-89) into one and rewrite them as follows.

 The primary objective of the present study is to identify the specific genetic locations and candidate genes associated with the development of lobed leaves in sorghum. Through this research, sblob1 (Sobic.003G010300) was suggested as a strong candidate gene accounting for the regulation of the lobed leaf phenotype observed in the sblob mutant. Moreover, this study also provide valuable insights into the regulatory processes accounting for lobed leaf formation in sorghum, thus supporting subsequent investigations in the field of sorghum functional genomics.

We agree that this will improve the flow and readability of the manuscript. Thank you for your suggestion.

Was the experiment performed with potted plants (Fig. 1A) or in the field (subsection 4.2)?

Response: The potted plants were only for the purpose of photography, as mentioned in the section 4.2. However, the plants were originally grown in the field before being transferred to pots for photographing. All of the other experiments were conducted in the field except for the grain weight. We have corrected the section 4.2 of the manuscript. We apologize for any confusion caused and will clarify this in the revised version of the manuscript.

L.123. “The assessment of chlorophyll levels...”. Analysis of chlorophyll content is not included in the Materials and Methods. Add please.

Response: Thank you for bringing this to our attention, and we sincerely apologize for the oversight. The chlorophyll content assessment has been meticulously incorporated into the Materials and Methods section, and the relevant text is now highlighted in red for easy identification:

 "To assess the chlorophyll content, fresh flag leaves from sblob and WT plants at the shooting and heading stages were harvested, weighed, and ground to a powder in liquid nitrogen for further analysis. The chlorophyll content was extracted using acetone and measured following a recently reported method[1].

  1. “the CYP71A1 gene codes ... a crucial component in the plant serotonin/melatonin biosynthesis pathway.” Why was the content of melatonin analyzed, and not serotonin, the synthesis of which is catalyzed by the product of the CYP71A1 gene (Fig. 4B)?

Response: We appreciate the thoughtful consideration given to our manuscript. We focus on analyzing melatonin rather than serotonin because of the roles these two compounds play in plant physiology. Serotonin serves as a precursor for melatonin synthesis, and melatonin consistently functions as the central molecule in the serotonin/melatonin biosynthesis pathway [2]. Extensive prior research indicates that serotonin does not play a predominant role in regulating biomass and leaf morphology. In contrast, numerous studies have demonstrated the significant impact of CYP71A1 on melatonin synthesis and its subsequent effects on biomass yield[3]. Therefore, our research focuses on melatonin to further investigate the regulatory role of CYP71A1 in biomass yield in sorghum.

We appreciate the reviewer's insightful query and hope this clarification provides a comprehensive rationale for our emphasis on melatonin in our study.

Thank you for your continued consideration.

L.243. “the content of melatonin and tryptamine...”. Tryptamine content analysis is not included in the Materials and Methods. Add please..

Response: In response to your suggestion, we have incorporated the necessary details for the analysis of tryptamine content using LC-MS/MS into the Materials and Methods section. The revised passage now reads as follows:

“4.10. Detection of tryptamine, melatonin and various hormones content

To further elucidate the physiological differences between the mutant plants and their corresponding wild-type counterparts, the levels of tryptamine, melatonin and various kinds of plant auxin in both plant types were measured. The flag leaves at heading stage were selected and harvested as the primary material for hormone analysis. To ensure accuracy, the sample of each biological replicate contained at least five flag leaves of individual plants. Each experiment contains at least two biological replicates. The detection of tryptamine, melatonin and auxin was conducted at MetWare (http://www.metware.cn/) based on the AB Sciex QTRAP 6500 LC-MS/MS platform which ensures accurate and reliable measurements[4].The extraction of those components was conducted based on the previous study[5]. The details of various kinds of auxin analyzed were listed in table S5. ”

We trust that this addition addresses the gap in our original submission, and we appreciate your diligence in ensuring the completeness of our manuscript.

Fig. 1. The leaf width of WT and sblob plants are almost the same in Fig. 1B, but differs by 1.5 times in Fig. 1D. Clarify please.

Response: We appreciate the opportunity to address the concern raised by the reviewer regarding the apparent discrepancy in leaf width measurements between Figures 1B and 1D.

We would like to clarify that the observed difference in leaf width between the WT and sblob plants in Figures 1B and 1D is not reflective of an actual biological variation but rather an artifact introduced during the imaging process. The phenomenon of water loss and leaf curling, often associated with detached leaves during imaging, can create visual disparities that may not accurately represent the true dimensions of the leaves.

It is important to note that our experimental data, including leaf width measurements, were meticulously obtained from in situ conditions within the field. The controlled environment of the field minimizes the impact of artifacts such as leaf curling due to water loss, ensuring the accuracy and reliability of our reported measurements. In the field, the biological variation could be easily detected (Figure S5).

Figure S5: the WT (left) and sblob (right) plants in the field. The picture was photographed on August 7th, 2023.We appreciate the reviewer's keen observation, and we would like to assure them and the editorial team that we are committed to maintaining the scientific integrity of our study. Thank you for your consideration and guidance through this review process.

Figure 6. Tryptamine is not melatonin or a hormone. The title of the figure should be corrected.

Response: The revised title for Figure 6 is as follows:

Figure 6. Melatonin, tryptamine and hormone content determination.

We believe this modification accurately reflects the components analyzed in the figure, and we are grateful for your attention to detail.

Subsection 4.10. The hormones analyzed and the sampling and analysis methods should be specified.

Response: In this section, we detail the materials, methods, and hormones analyzed to investigate the physiological distinctions between mutant and wild-type plants. The analysis focused on flag leaves at the heading stage as the primary plant material.  Subsection 4.10 was rewritten as follows:

“4.10. Detection of tryptamine, melatonin and various hormones content

To further elucidate the physiological differences between the mutant plants and their corresponding wild-type counterparts, the levels of tryptamine, melatonin and various kinds of plant auxin in both plant types were measured. The flag leaves at heading stage were selected and harvested as the primary material for hormone analysis. To ensure accuracy, the sample of each biological replicate contained at least five flag leaves of individual plants. Each experiment contains at least two biological replicates. The detection of tryptamine, melatonin and auxin was conducted at MetWare (http://www.metware.cn/) based on the AB Sciex QTRAP 6500 LC-MS/MS platform which ensures accurate and reliable measurements[4].The extraction of those components was conducted based on the previous study[5]. The details of various kinds of auxin analyzed were listed in table S5. ”

The technical details of the LC-MS/MS method are shown below to further clarify the reviewers’ question, but this section is not reflected in the manuscript:

  1. Chemicals and reagents

HPLC grade acetonitrile (ACN) and methanol (MeOH) were purchased from Merck (Darmstadt, Germany). MilliQ water (Millipore, Bradford, USA) was used in all experiments. All of the standards were purchased from Olchemim Ltd. (Olomouc, Czech Republic) and isoReag (Shanghai, China). Acetic acid and formic acid were bought from Sigma-Aldrich (St Louis, MO, USA). The stock solutions of standards were prepared at the concentration of 1 mg/mL in MeOH. All stock solutions were stored at -20°C. The stock solutions were diluted with MeOH to working solutions before analysis[5].

  1. Sample preparation and extraction

Fresh plant sample was harvested, immediately frozen in liquid nitrogen, ground into powder (30 Hz, 1 min), and stored at -80°C until needed. 50 mg of plant sample was weighed into a 2 mL plastic microtube and frozen in liquid nitrogen, dissolved in 1 mL methanol/water/formic acid (15:4:1, V/V/V). 10 μL internal standard mixed solution (100 ng/mL) was added into the extract as internal standards (IS) for the quantication. The mixture was vortexed for 10 minutes, then centrifugation for 5 min (12000 r/min, and 4°C), the supernatant was transferred to clean plastic microtubes, followed by evaporation to dryness and dissolved in 100 μL 80% methanol (V/V), and filtered through a 0.22 μm membrane filter for further LC-MS/MS analysis.

We believe that these additional details offer clarity regarding the experimental procedures undertaken in this study. We appreciate the reviewer's guidance and remain dedicated to upholding the accuracy and completeness of our manuscript.

Fig. S3, S4. Please indicate the differences between the right and left parts of the figures.

Response: Fig. S3 and S4 have been corrected based on your advice. The red box in the Fig. S3 and S4 indicated the differences between the two biologic replicated samples. We believe that your suggestions could help to uphold the accuracy and completeness of our manuscript.

Fig. S4. The title of the figure is written as S3. Correct please.

Response: The mistake has been corrected and we are grateful for your attention to detail.

Table S1. “uesd” should be corrected to “used”.

Response: The mistake has been corrected and we apologize for our carelessness and we are grateful for your attention to detail.

References:

  1. Wang, Y.; Wang, J.; Chen, L.; Meng, X.; Zhen, X.; Liang, Y.; Han, Y.; Li, H.; Zhang, B. Identification and function analysis of yellow-leaf mutant (YX-yl) of broomcorn millet. BMC Plant Biology 2022, 22, 463, doi:10.1186/s12870-022-03843-y.
  2. Bhowal, B.; Bhattacharjee, A.; Goswami, K.; Sanan-Mishra, N.; Singla-Pareek, S.L.; Kaur, C.; Sopory, S. Serotonin and Melatonin Biosynthesis in Plants: Genome-Wide Identification of the Genes and Their Expression Reveal a Conserved Role in Stress and Development. Int J Mol Sci 2021, 22, doi:10.3390/ijms222011034.
  3. Nawaz, K.; Chaudhary, R.; Sarwar, A.; Ahmad, B.; Gul, A.; Hano, C.; Abbasi, B.H.; Anjum, S. Melatonin as Master Regulator in Plant Growth, Development and Stress Alleviator for Sustainable Agricultural Production: Current Status and Future Perspectives. Sustainability 2021, 13, 294.
  4. Xu, X.; Murphy, L.A. Fast and sensitive LC-MS/MS method for quantification of cannabinoids and their metabolites in plasma of cattle fed hemp. Journal of Separation Science n/a, 2300630, doi:doi.org/10.1002/jssc.202300630.
  5. Li, Y.; Zhou, C.; Yan, X.; Zhang, J.; Xu, J. Simultaneous analysis of ten phytohormones in Sargassum horneri by high-performance liquid chromatography with electrospray ionization tandem mass spectrometry. J Sep Sci 2016, 39, 1804-1813, doi:10.1002/jssc.201501239.

Round 2

Reviewer 1 Report

Comments and Suggestions for Authors

Manuscript can be accepted for pubblication.